# Automatic Early Warning to Derive Eruption Source Parameters of Paroxysmal Activity at Mt. Etna (Italy)

**Luigi Mereu** [1,2,*], **Frank Silvio Marzano** [2,3,†], **Costanza Bonadonna** [4], **Giorgio Lacanna** [5], **Maurizio Ripepe** [5] **and Simona Scollo** [6]

1    Istituto Nazionale di Geofisica e Vulcanologia, Sezione di Bologna, 40127 Bologna, Italy
2    Center of Excellence CETEMPS, University of L'Aquila, 67100 L'Aquila, Italy
3    Dipartimento di Ingegneria dell'Informazione (DIET), Sapienza University of Rome, 00184 Rome, Italy
4    Department of Earth Sciences, University of Geneva, 1205 Geneva, Switzerland; costanza.bonadonna@unige.ch
5    Department of Earth Science, University of Florence, 50121 Florence, Italy; giorgio.lacanna@unifi.it (G.L.); maurizio.ripepe@unifi.it (M.R.)
6    Istituto Nazionale di Geofisica e Vulcanologia, Osservatorio Etneo, 95125 Catania, Italy; simona.scollo@ingv.it
*    Correspondence: luigi.mereu@ingv.it
†    Deceased.

**Abstract:** Tephra dispersal and fallout resulting from explosive activity of Mt. Etna (Italy) represent a significant threat to the surrounding inhabited areas as well as to aviation operations. An early-warning system aimed at foreseeing the onset of paroxysmal activity has been developed, combining a thermal infrared camera, infrasonic network, and a weather radar. In this way, it is possible to identify the onset of a lava fountain as well as to determine the associated mass eruption rate (MER) and top plume height ($H_{TP}$). The new methodology, defined as the paroxysmal early-warning (PEW) alert system, is based on the analysis of some explosive eruptions that occurred between 2011 and 2021 at Etna, simultaneously observed by the thermal camera and infrasound systems dislocated around the summit eruptive craters, and by the weather radar, located at about 32 km from the summit craters. This work represents an important step towards the mitigation of the potential impact associated with the tephra dispersal and fallout during paroxysms at Etna, which can be applied to other volcanoes with similar activity and monitoring strategies.

**Keywords:** infrasonic pressure; microwave radar observables; thermal infrared camera image; paroxysmal activity; mass eruption rate; exit velocity; paroxysmal early warning





## 1. Introduction

Tephra injected into the atmosphere during explosive eruptions poses a direct threat to aviation and can have primary and secondary impacts at different spatial and temporal scales on surrounding communities [1–3]. Current monitoring procedures necessary to mitigate the potential impact of tephra dispersal and fallout are mostly expert-visioned, resulting into a delay ranging from a few minutes to a few hours from the occurrence of an event [4]. Automatic systems for the detection of explosive activity based on volcanic tremor [5] and infrasound [6,7] are essential for early warning, but they do not provide eruptive source parameters (ESPs), such as the top plume height, size, and the mass and/or volume eruption rate at the source. Infrasound arrays can be used to detect the onset of the eruption in near-real-time up to distances of a few hundreds of kilometers. The infrasonic signals are acoustic waves produced by explosive gas expansion (mainly water vapor) in the atmosphere, with frequencies less than 20 Hz. The analysis of this signal is fundamental for both volcano monitoring purposes and for a better understanding of eruption dynamics [8,9]. Volcano observatories around the world routinely use infrasound networks aimed at detecting, locating, and characterizing volcanic activity. These systems

have shown promising results in forecasting paroxysmal activity at open-vent systems and providing new insights on eruption dynamics and volcanic processes [10,11]. An automatic warning system of explosive eruptions, based on the infrasound system (IS), has also been developed at Etna [6,12,13]. The infrasonic observations are collected by two small-aperture arrays, showing how the eruptive episodes are marked by a smooth increase in amplitude preceding the proper paroxysmal phase, sometimes also hours before the formation of a volcanic plume. Today, the link between acoustic pressure and eruptive dynamics of Strombolian eruptions at the Etna volcano is well-understood [8], as well as the correlation between the eruptive phase and other sensors' measurements [14,15]. Thermal cameras (TC) are usually employed to monitor the explosive activities generating lava fountains [10,16,17]. Indeed, the analysis of these images allows to identify the lava fountains as the bi-dimensional temperature-saturated region, named the incandescent jet region (IJR) [18]. The height of the IJR can be converted into exit velocities of pyroclastic material at the volcanic vent and, in turn, into the erupted volume [16,19]. TC can work both in good weather conditions and during cloudy days, albeit with reduced sensitivity. The weather radar (WR) has shown great potential in evaluating real-time features of the explosive activity [20]. In fact, WR visibility is not hindered by weather clouds. Applying the Volcanic Ash Radar Retrieval (VARR) approach [14,21,22] to WR observables makes it possible to quantify the main ESPs in near-real-time [23–25]. The early-warning system proposed in this work combines the ability of infrasound and the thermal camera in detecting and timing the onset of the explosive activity with that of the weather radar to probe the volcanic plume and retrieve key ESPs, e.g., the mass eruption rate and the plume height [14]. We have developed a new algorithm, named paroxysmal early warning (PEW), based on the combination of IS, TC, and WR. The method was tested on nine cases of Etna lava fountains: (1) 10 April 2011, (2) 12 April 2012, (3) 23 November 2013, (4) 3–5 December 2015, and (5) 16 and 23 February 2021. The work is organized as follows: Etna paroxysmal activities are briefly described in Section 2, the methodology for each sensor is presented in Section 3, the applications to several Etna explosive events are discussed in Section 4, the combined approach for PEW is presented in Section 5, and the discussion and concluding remarks are provided in Section 6.

## 2. Etna Paroxysmal Activity

Etna is one of the most active volcanoes in Europe, located on the east coast of Sicily (Italy). Etna comprises four main craters: Voragine (VOR), Bocca Nuova (BN), the Northeast crater (NE), and the Southeast crater (SE) (Figure 1a). Among the typical Etna activities, we observed the paroxysmal events which generate tephra-rich volcanic plumes [17,26,27]. Over the last decade, Etna has produced many paroxysms that are characterized by volcanic emissions with abundant coarse-grained tephra falling in the proximal area and fine ash that can be dispersed hundreds of kilometers [28–30]. As usual, at Etna, the dynamics of explosive basaltic eruptions switch from Strombolian activity, denoting frequent, small-scale, transient explosions [31], to the paroxysmal phase, taking place with an impulsive trend and characterized by rapid waxing and waning phases [16].

During lava fountains, the formation of IJR occurs, and it is related to a different mechanism within the conduit. The eruptive column is generally of a few km, while the dispersed volcanic plume could exceed 15 km of altitude above sea level [4,5,25,30,32,33].

In this paper, we focus our attention on nine paroxysmal events. On 10 April 2011, the paroxysm was weak, with a plume up to 7 km above sea level (a.s.l.) [29]. Similar behavior was observed on 12 April 2012, which generated a tephra plume of a few km a.s.l. On 23 November 2013, sustained lava fountains generated a tephra plume of 11–12 km a.s.l. [26,29]. Starting from the early morning of 3 December to the afternoon of 5 December 2015, four paroxysms were produced from the VOR crater, with tephra plumes up to 12–16 km a.s.l. [24]. We analyzed the recent lava fountain produced on 16 and 23 February 2021, which formed high tephra plumes from the SE crater of about 10–12 km a.s.l. [19].

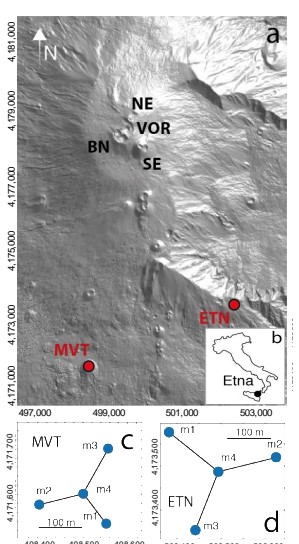
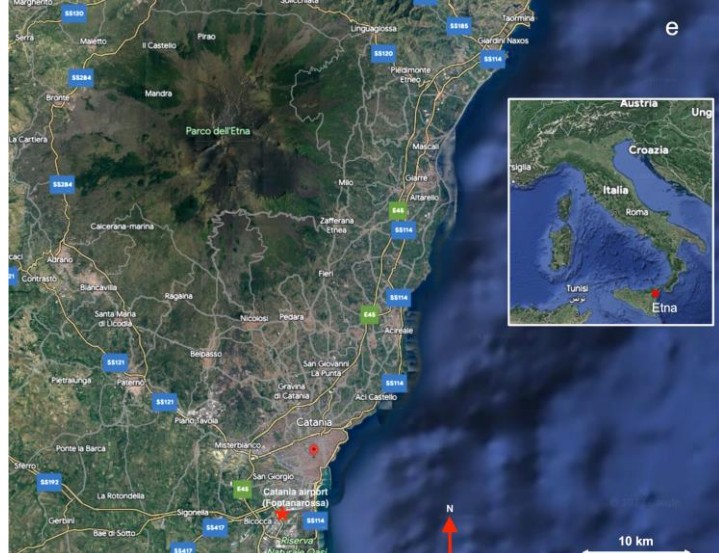

**Figure 1.** (**a**) The DEM of Etna located in southern Italy, showing the summit craters (Southeast crater, SE; Bocca Nuova, BN; Voragine, VOR; Northeast crater, NE) and the locations of the two infrasound arrays (ETN, MVT), from [6]. (**b**) Etna location in Sicily, Italy. Infrasonic arrays at Etna with a triangular geometry and a maximum aperture of 150 m for the MVT array and 250 m for the ETN array, panels are shown in (**c**,**d**), where m1, m2, m3, and m4 identify respective differential pressure transducers. (**e**) A map of the area, which includes all the sensors used in this work.

## 3. Early-Warning Systems

We identified three early-warning (EW) phases, named EW0 (quiescent or eruptive phase with sporadic ash emission), EW1 (onset of the Strombolian activity), and EW2 (onset of the lava fountain activity and associated tephra plume). This threshold methodology can be assimilated to a Bayesian approach, in which the probabilities of having certain EW levels are interpreted as confidence intervals for which a specific event/condition occurs.

### 3.1. Infrasonic System (IS)

The IS consists of two devices (Figure 1a,b): one infrasonic array, ETN, was deployed in September 2007 on the southern edge of the Valle del Bove, at an elevation of 2100 m a.s.l. and 5500 m from the summit craters; the second array, MVT, was installed in 2012 in the forested area of Monte Vetore, on the south flank of Etna volcano at an elevation of 1800 m a.s.l., about 6500 m from the summit craters. Both arrays have a triangular geometry (Figure 1c for MVT and Figure 1d for ETN) and are equipped with four differential pressure transducers (all sensors—1 MBAR-D-4V), with a sensitivity of 25 mV/Pa, able to detect atmospheric pressure fluctuations in the range of +/− 100 Pa. Sensors have a flat response between 0.01 Hz and 20 Hz and are installed within waterproof cases buried ~1 m deep. For both arrays, the minimum detectable pressure is approximately $10^{-2}$ Pa. The limited power requirement of the array (<1 W) and the use of fiber optic has allowed, for the last 10 years, an operative efficiency of >95% [32]. Both arrays are displaced in the Etna summit area, as highlighted in Figure 1e, about 30 km from Catania city.

For all detections having values of back-azimuth and apparent velocity consistent with the position of the summit craters, we estimated an infrasonic parameter (IP), obtained as the product between the mean infrasonic amplitude ($A_P$) and the number of detections ($N_D$) per minute (IP = $A_P \times N_D$), as described in [6]. The IP is strongly related to the persistence of the infrasound signal, and here, it is used to derive the EW.

The efficiency of the IS in the evaluation of EW has already been tested for several explosive activities of Etna [6]. When the explosive activity increases, the IP increases with respect to the value of −1, which occurs when in one minute there is no infrasound data detection, and this allows to identify the onset of the Strombolian activity. The

infrasonic system-early warning (IS-EW) operates on two IP thresholds that correspond to the beginning of the precursory violent Strombolian phase (EW1), when IP > 60 for more than 5 min (orange in Figure 2), and to the onset of the lava fountain (EW2), when IP > 120 for more than 5 min (red in Figure 2). These two thresholds were used to define two alert levels in the EW procedure and to deliver automatic notifications. The latency of the automatic notification for lava fountains at Etna was below 2 min [6].

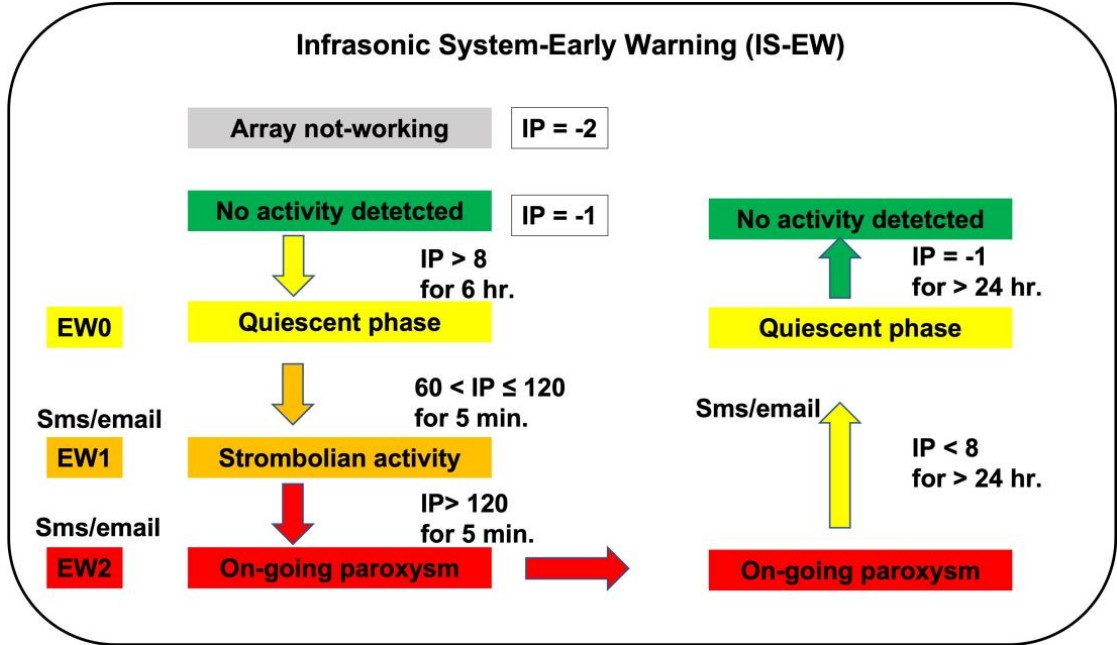

**Figure 2.** IS-EW flowchart showing the relation among the infrasound parameter (IP) (dimensionless) and corresponding alert levels (EW0 in yellow, EW1 in orange, and EW2 in red) (modified from [6,15]).

### 3.2. Thermal Camera (TC)

The video-monitoring network system of the Istituto Nazionale di Geofisica e Vulcanologia, Osservatorio Etneo (INGV-OE), consists of the ENT camera, located in Nicolosi, about 15 km south of the Etna summit craters, and belonging to the Etna video-surveillance system. The ENT camera is equipped with an A40M Thermovision Forward-Looking InfraRed (FLIR Systems) camera, and records in the 7.5 and 13 μm spectral range, providing a time series of 640 × 480-pixel images with a spatial resolution of 1.3 μrad (few meters) and a thermal sensitivity of 80 mK at 25 °C. Thermal images of ENT are displayed with a fixed color scale corresponding to the temperature range from −20 to 80 °C [34].

Images recorded by ENT provide the brightness temperature over the whole eruptive episode. Employing the algorithm described in [18], we detected the saturated region evaluating the brightness temperature gradient as a function of a given threshold, to detect the height of the IJR. As a first approximation, the height ($H_{IJR}$) of the IJR can be considered as the height of the lava fountain. The exit velocity, $v_{ex}$ (m/s), of pyroclastic material is derived from the $H_{IJR}$, starting from the Torricelli relation, assuming that: (i) most of the pyroclasts are sufficiently large, which can be considered as accelerated projectiles confined only in this incandescent jet region, and (ii) atmospheric density variations and drag effects are negligible. Under these assumptions, at each timestep, $t$, we applied the relation: $v_{ex}(t) = (2 \times g \times H_{IJR}(t))^{0.5}$, where $g$ (m/s$^2$) is the earth gravity acceleration [18].

The thermal camera-early warning (TC-EW) follows the same flowchart previously described for the IP signal (Figure 3).

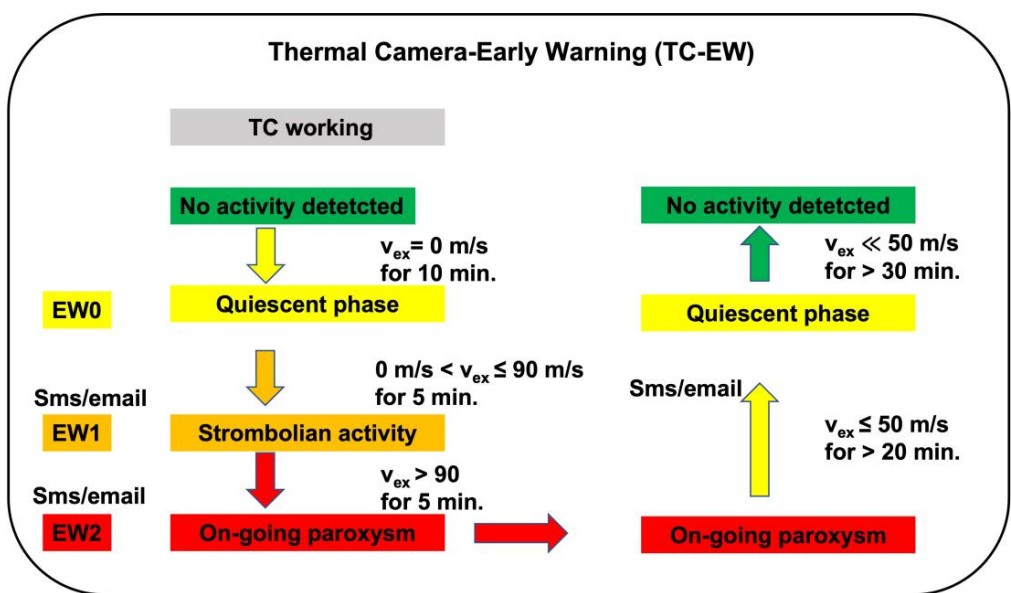

**Figure 3.** TC-EW flowchart showing the relation between the exit velocity, $v_{ex}$ (m/s), of pyroclastic material, as derived by TC, and the corresponding alert levels (EW0 in yellow, EW1 in orange, and EW2 in red) (modified from [6]).

In this case, from the quiescent activity with $v_{ex}$ equal to 0 m/s (EW0, yellow box), the beginning of the precursory Strombolian activity, corresponding to EW1 (orange box), was delivered when the $v_{ex}$ was greater than 0 m/s and lower than 90 m/s. When $v_{ex}$ exceeded 90 m/s for more than 5 min, we identified the onset of the explosive eruption, EW2 (red box).

### 3.3. Weather Radar (WR)

Radar technology is well-established and can nowadays provide fast 3D volume scanning together with Doppler and dual-polarization capabilities [35,36]. The detection sensitivity of the WR depends on several factors, such as the distance between the radar antenna and the target, the transmitter central wavelength, the receiver minimum detectable power, and the resolution volume [20,23].

In this work, we used the observables of the X-band WR, working at 9.6 GHz, permanently positioned at the Catania airport (Fontanarossa), approximately 32 km from Etna. This active sensor allows to probe the surrounding scene as a function of range (~160 km), azimuth (360°), and antenna-pointing elevation, with respect to the horizon covering a height of 20 km. The whole volume acquisition time was 10 min, with a radial resolution of 200 m [37].

Applying the VARR algorithm to radar data, we estimated several ESPs. In particular, we determined the mass eruption rate ($Q_M$) based on an evaluation of the space–time variation of the pyroclastic concentration in the whole volume above the eruptive crater scanned by the WR [18,22–25]. It is worth noting that the WR extension to the automatic detection of the volcanic plume within an EW system for volcanic tephra hazard forecasting remains an open question [14].

The eruption source parameters associated with the weather radar are named as WR-ESPsR (Figure 4). We used a similar scheme as for the two previous systems, but with thresholds consistently adapted to operate on the $Q_M$ retrievals. The alert $Q_M$-*low* was delivered when $Q_M$ was between $10^5$ and $10^6$ kg/s for more than 10 min (orange in Figure 4), whereas $Q_M$-*high* was delivered when $Q_M > 10^6$ kg/s for more than 10 min.

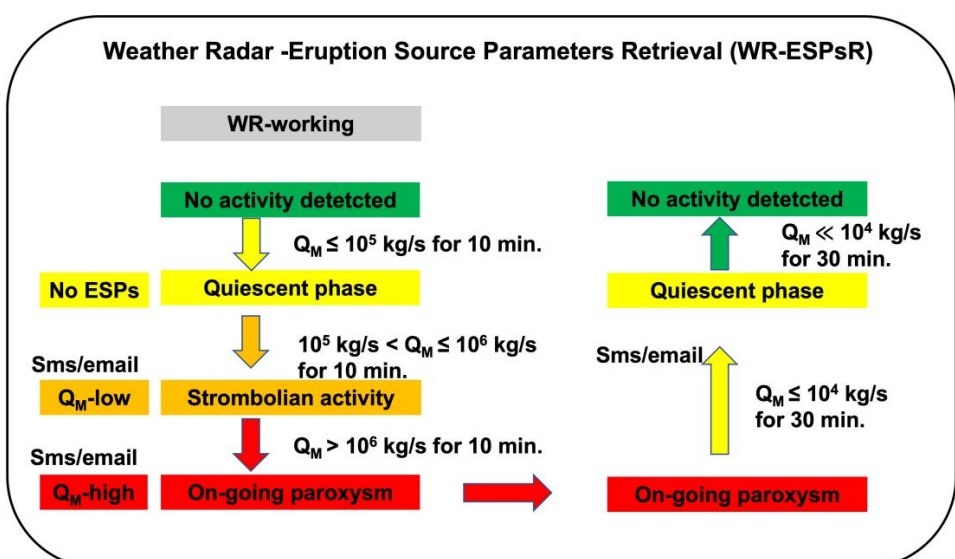

**Figure 4.** The WR-ESPsR flowchart: relations among $Q_M$ (kg/s) retrievals derived from the WR and corresponding $Q_M$ levels (low values in orange and high values in red) are shown.

## 4. Application to Etna

We have quantitatively applied the PEW scheme to different Etna lava fountains that occurred between 2011 and 2021, for which data from the three systems were available. In Figure 5, we show the RGB frames recorded by the ENT camera in Nicolosi for: (a) 10 April 2011 at 10:32, (b) 12 April 2012 at 14:57, (c) 23 November 2013 at 10:05, (d) 3 December 2015 at 02:49, (e–f) 4 December 2015 at 09:24 and 20:33, respectively, (g) 5 December 2015 at 14:55, (h) 16 February 2021 at 16:33, and (i) 23 February 2021 at 23:48.

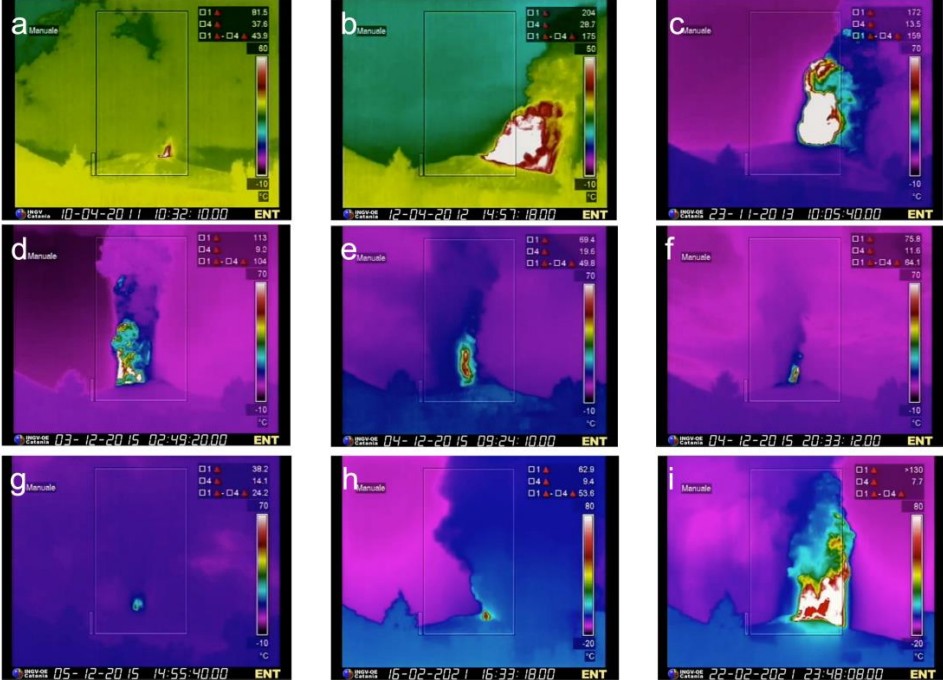

**Figure 5.** The original RGB thermal images from the ENT camera for each Etna event (**a–i**) as specified in the black bar below: the date (dd-mm-yyyy) and UTC time (hh:mm:ss:00). The color bar on the right side of each picture shows the uncalibrated temperature ranging between $-10/-20$ °C and $50/80$ °C.

It should be noted that this system has only been tested and applied to a limited number of events selected as test cases due to the availability of simultaneous sensor measurements (WR, TC, and IS). For each eruptive episode, we evaluated the correlation among the $Q_M$ retrieval (solid blue line), the IP (dimensionless) signal (dashed magenta line), and the $v_{ex}$ estimation (solid magenta line) to prove the sensitivity of possible real-time automatic system implementation. Each EW level, associated with a specific sensor, is plotted as dashed lines with different colors: WR-$Q_M$L (light blue dashed line), WR-$Q_M$H (blue dashed line), IS-EW1 (orange dashed line), IS-EW2 (red dashed line), TC-EW1 (green dashed line), and TC-EW2 (dark dashed line).

### 4.1. 10 April 2011

For this event, the TC and IS anticipated the WR detection of the lava fountain more than two hours before (Figure 6), as confirmed by the associated EW1 levels. The IP signal showed a swinging variability between 100 and about 300, whereas the $v_{ex}$ was about 90/100 m/s in the morning, but reached a peak value of about 160 m/s in the late afternoon.

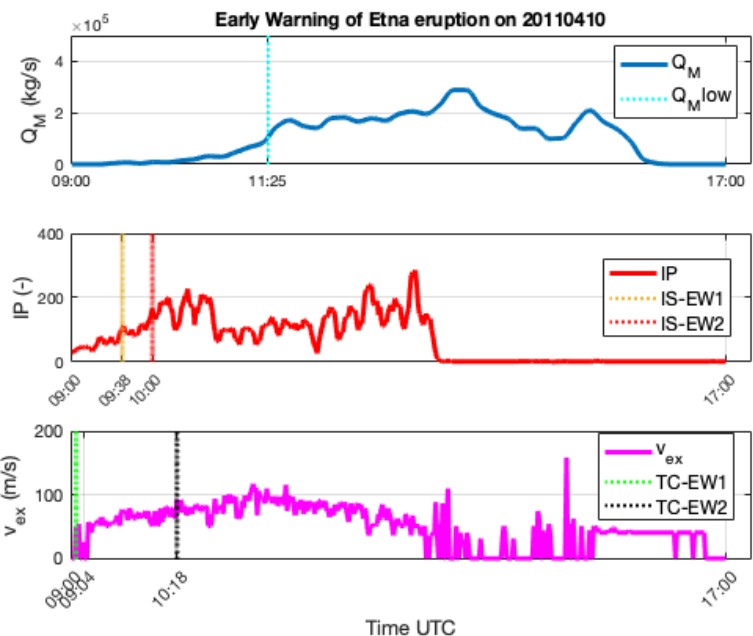

**Figure 6.** Time series of $Q_M$ (kg/s), IP (-), and $v_{ex}$ (m/s) in blue, red, and magenta lines, respectively, for the 10 April 2011 Etna lava fountain. Values are plotted together with the EW1 and EW2 (vertical dashed lines of different colors).

The lava fountains were weak and prolonged, generating a volcanic plume of about 8.5–9 km a.s.l. and observed by the WR when the IP normalized (IP < 8), around 13:00 UTC. The explosive activity was weak ($Q_M \sim 2 \times 10^5$ kg/s) and long-lasting (about 5 h). The TC detected the EW1 level at 09:04, about half an hour before the IP, while the EW2 level was detected a few minutes later than that observed by the IP.

### 4.2. 12 April 2012

On 12 April 2012, the volcanic plume detected by WR was foretold by almost three hours and two and a half hours, by IS and TC, respectively (Figure 7). This activity started in the evening of the day before as Strombolian activity, evolving to a lava fountain in the early afternoon of April 12. The fountaining was concentrated within an hour, in which the three systems presented a rapid and simultaneous rise of the estimated values. It was interesting to observe that: (1) both EW1 and EW2 of TC anticipated the IS values, highlighting a prolonged Strombolian activity, (2) both the EW levels of WR were quite close in time, and (3) all three systems showed a simultaneous decrease in the estimated values.

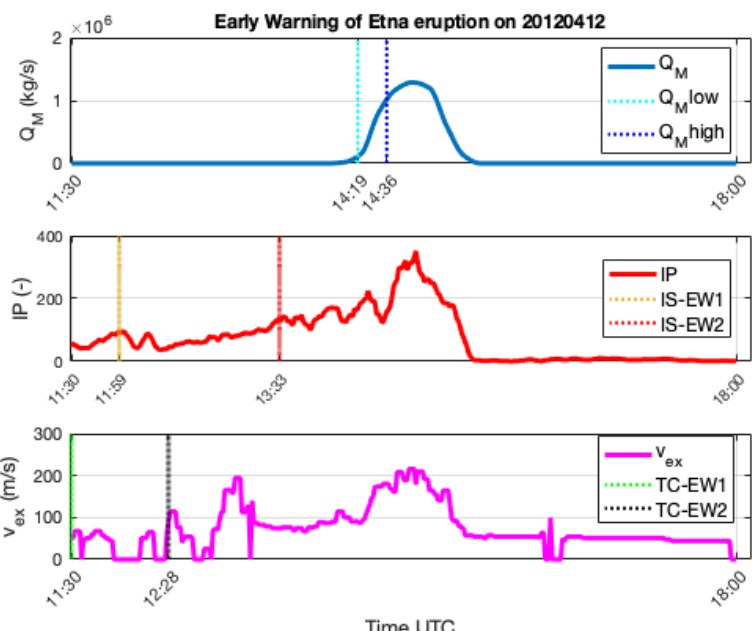

**Figure 7.** In each panel, the time series of $Q_M$ (kg/s), IP (-), and $v_{ex}$ (m/s) are shown as blue, red, and magenta lines, respectively, for the 12 April 2012 Etna lava fountain. The EW level time steps are plotted as vertical dashed lines with different colors, identifying the associated sensors.

### 4.3. 23 November 2013

The TC-EW1 anticipated both the IS-EW levels about half an early, whereas the TC-EW2 was a precursor of both WR-$Q_M$ levels. The $Q_M$ reached a maximum value of about $4.8 \times 10^6$ kg/s, simultaneous with the increase of the $v_{ex}$ derived from the TC (Figure 8). It is worth noting that the maximum $Q_M$ was detected almost 15 min after the maximum peak of the IP signal. Additionally, in this case, the TC and WR systems detected the lava fountain about one hour before.

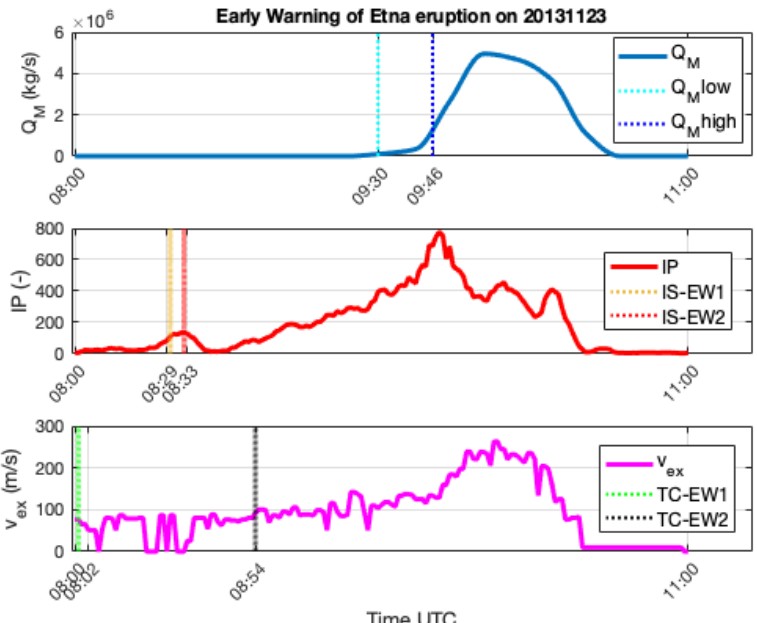

**Figure 8.** Time series of $Q_M$ (kg/s), IP (-), and $v_{ex}$ (m/s) are shown as blue, red, and magenta lines, respectively, for the 23 November 2013 Etna lava fountain. Values are plotted together with the EW1 and EW2 (vertical dashed lines with different colors).

*4.4. 3–5 December 2015*

Four lava fountain episodes with high volcanic plumes occurred between 3 and 5 December 2015. On 3 December, the EW levels of TC preceded both the IS and WR detection, allowing to consider the TC system as a valid pre-alert for explosive events (Figure 9). TC-EW2 was detected more than half an hour before the growth of the IP signal. We observed an advance of the $v_{ex}$ respect to the $Q_M$ of about 30 min. The maximum value of $Q_M$ during the paroxysm was about $5 \times 10^6$ kg/s, while the $v_{ex}$ reached 200 m/s.

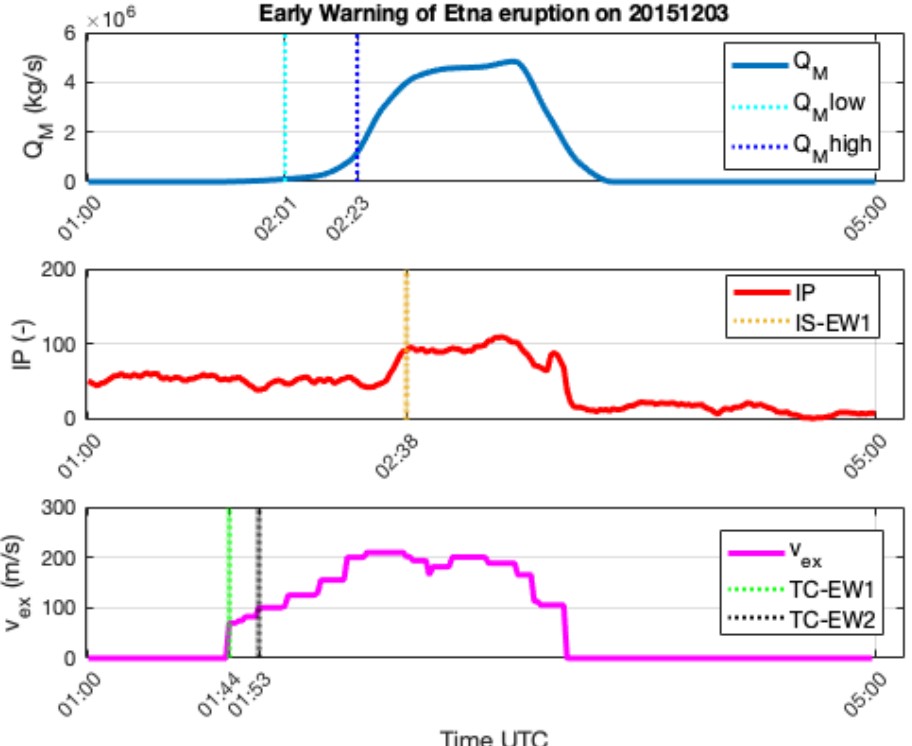

**Figure 9.** Time series of $Q_M$ (kg/s), IP (-), and $v_{ex}$ (m/s) are shown as blue, red, and magenta lines, respectively, for the 3 December 2015 Etna lava fountain. The EW level time steps are plotted as vertical dashed lines with different colors, identifying the associated sensors.

On the morning of 4 December, the IS-EW1 (Figure 10) anticipated the lava fountain about two hours before, which lasted just over an hour (09:00 to 10:15 UTC) and was indicated by the EW levels of the three sensors in very close time windows. The $v_{ex}$ and $Q_M$ were almost overlapping over time, whereas the IP signal had not yet reached its maximum value over 250. In this case, the rapid growth of $v_{ex}$ allowed directly defining the TC-EW2 (which coincides with TC-EW1). The IS signal lasted about 30 min longer than the WR and TC signals.

On the evening of 4 December (Figure 11), the IS-EW1 alert remained for the whole day, although a slight increase of the IP signal was observed. The EW2 level was detected at 20:30 UTC, reaching a value of about 300 at 21:30 UTC. The WR-$Q_M$ high and EW2 levels from TC were observed in correspondence with the growth of the associated estimates of $v_{ex}$ and $Q_M$, achieving maximum values of 70 m/s and about $1.6 \times 10^6$ kg/s, respectively. The trends of both $Q_M$ and $v_{ex}$ did not show a similar time extension. In fact, the $v_{ex}$ was limited to a smaller time interval than that of $Q_M$. Additionally, in this case, TC anticipated the EW levels of the other sensors.

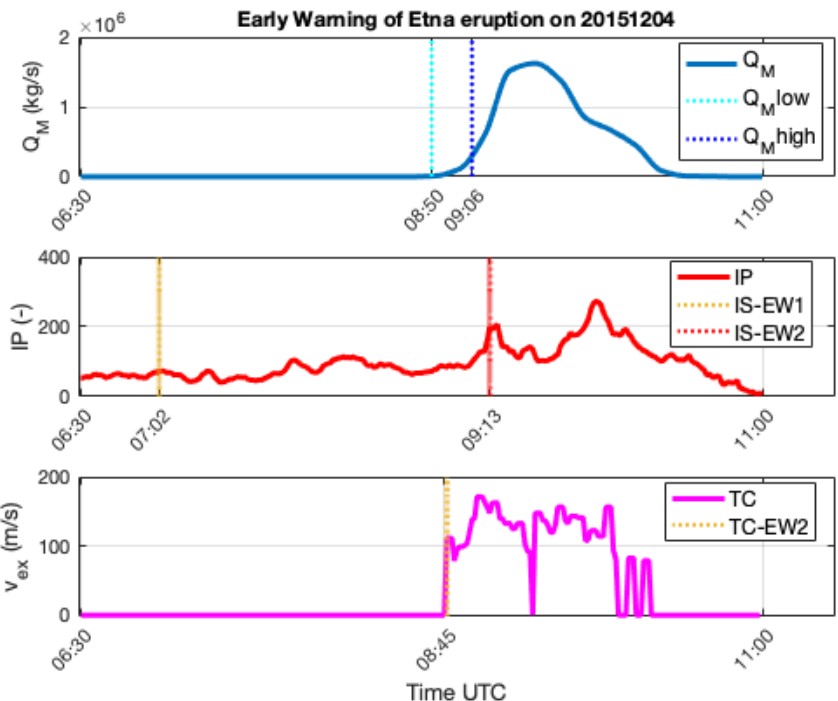

**Figure 10.** Time series of $Q_M$ (kg/s), IP (-), and $v_{ex}$ (m/s) shown in blue, red, and magenta lines, respectively, for the morning of the 4 December 2015 Etna lava fountain. Values are plotted together with the EW1 and EW2 (vertical dashed lines with different colors).

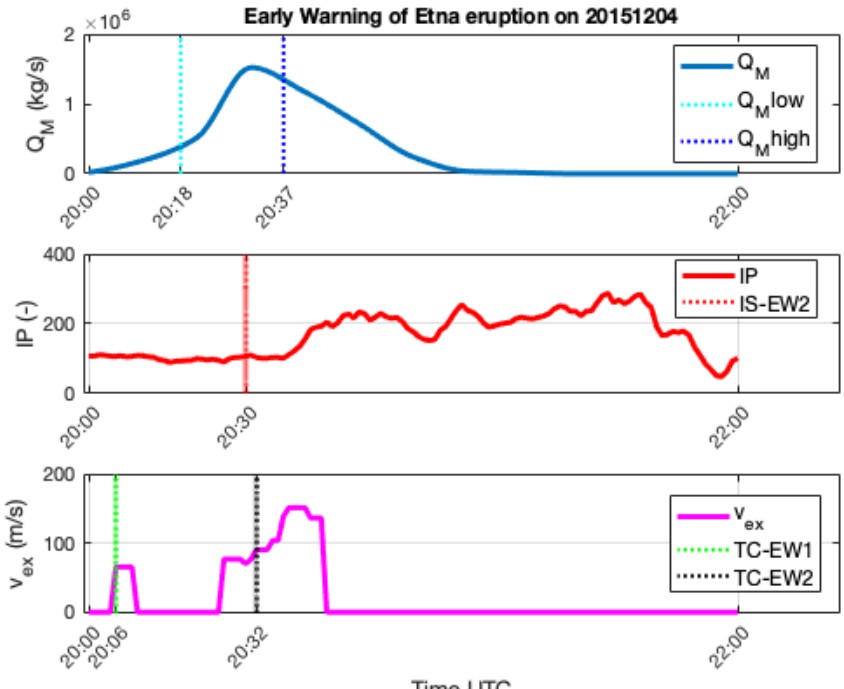

**Figure 11.** Time series of $Q_M$ (kg/s), IP (-), and $v_{ex}$ (m/s) are shown as blue, red, and magenta lines, respectively, for the evening of 4 December 2015 Etna lava fountain. The EW level time steps are plotted as vertical dashed lines with different colors, identifying the associated sensors.

The event on 5 December 2015 (Figure 12) was that in which the TC system was still confirmed as a good precursor. The misalignment between the WR and TC signals' is less than 30 min, also confirmed by the associated EW levels and the $Q_M$ level. This difference was probably due to the contamination of the scene by the presence of meteorological

clouds that prevented the view of the summit crater from the TC. Both EW1 of WR and IS were identified very close in time. The $v_{ex}$ trend reached a maximum value of about 160 m/s, but it was limited in time (14:20 to 15:10 UTC) with respect to $Q_M$, which showed a value of about $0.8 \times 10^6$ kg/s and a longer time interval (14:20 to 17:20 UTC). The IP signal showed values lower than 200 during most of the event.

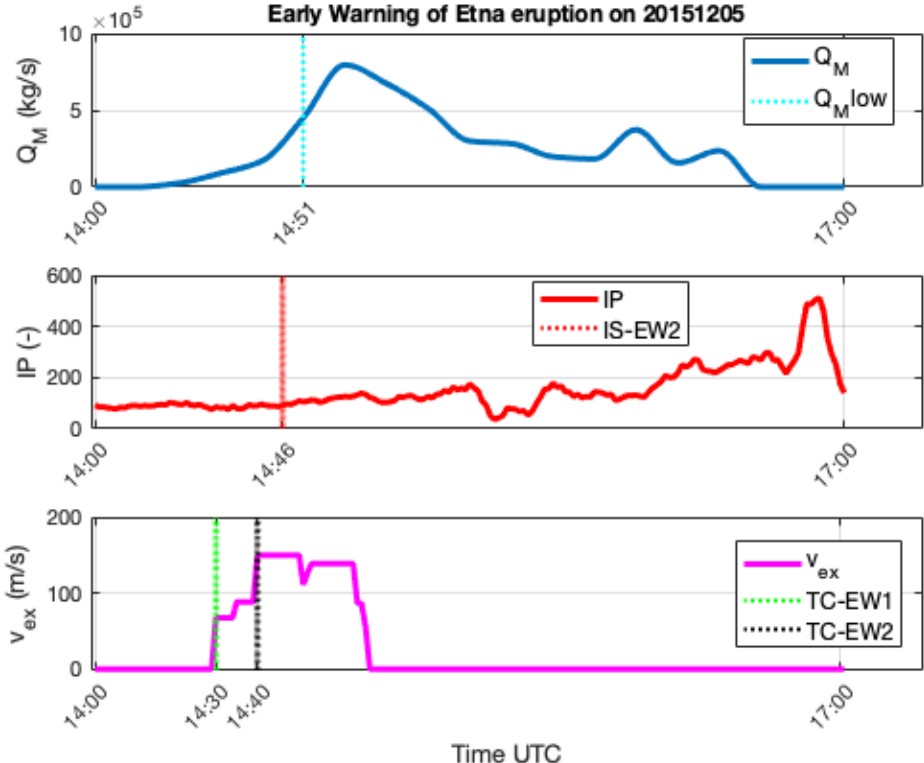

**Figure 12.** Time series of $Q_M$ (kg/s), IP (-), and $v_{ex}$ (m/s) are shown as blue, red, and magenta lines, respectively, for the 5 December 2015 Etna lava fountain. The EW level time steps are plotted as vertical dashed lines with different colors, identifying the associated sensors.

*4.5. 16 and 23 February 2021*

We focused our analysis on two episodes in the middle of February 2021. The first was related to February 16 (Figure 13), where we observed that the TC-EW1 preceded the WR and TC detections. In fact, the IS-EW1 and IS-EW2 signals have anticipated of about 20 min the WR-EW1 signal. It is interesting to note that the IP shows a peak of 310 more than one hour after the maximum value of $7 \times 10^5$ kg/s reached by $Q_M$. On the contrary, from the TC images, it was not possible to estimate significant $v_{ex}$ values because of a partial occlusion of the TC field of view, due to the accumulation of volcanic ash around the camera.

Figure 14 shows the signals for the lava fountain of 23 February 2021. The $Q_M$, $v_{ex}$, and IP values reached values of $2 \times 10^6$ kg/s, 220 m/s, and about 200, respectively, between 23:30 and 00:00 UTC. It is interesting to note the good overlapping, confirming the possible and reliable integration of these systems in the definition of the automated paroxysmal early warning. The TC-W1 generally anticipated the EW levels of the other two sensors, while the WR-EW levels were mainly close together in time as the $Q_M$ increased.

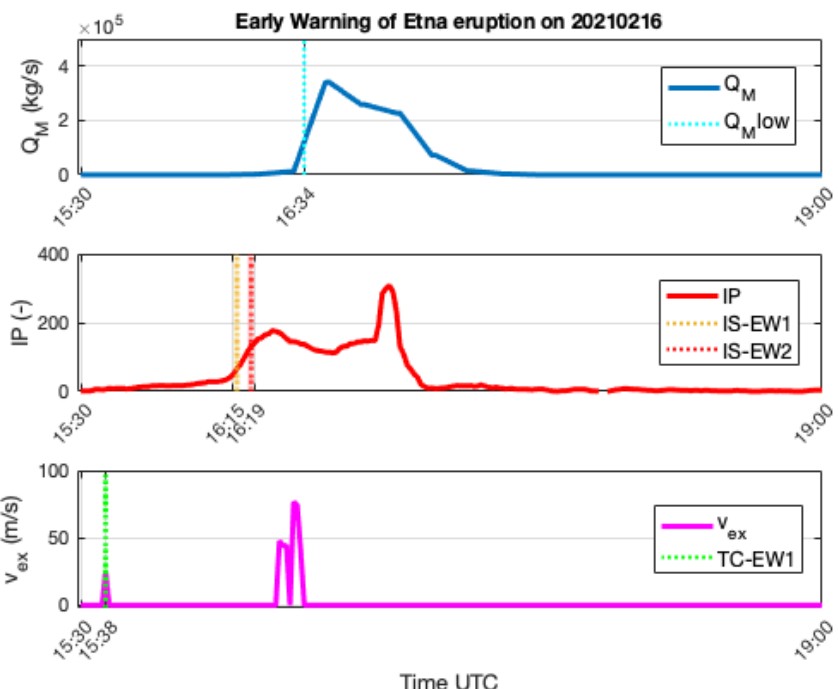

**Figure 13.** Time series of $Q_M$ (kg/s), IP (-), and $v_{ex}$ (m/s) shown as blue, red, and magenta lines, respectively, for the 16 February 2021 Etna lava fountain. Values are plotted together with the EW1 and EW2 (vertical dashed lines with different colors).

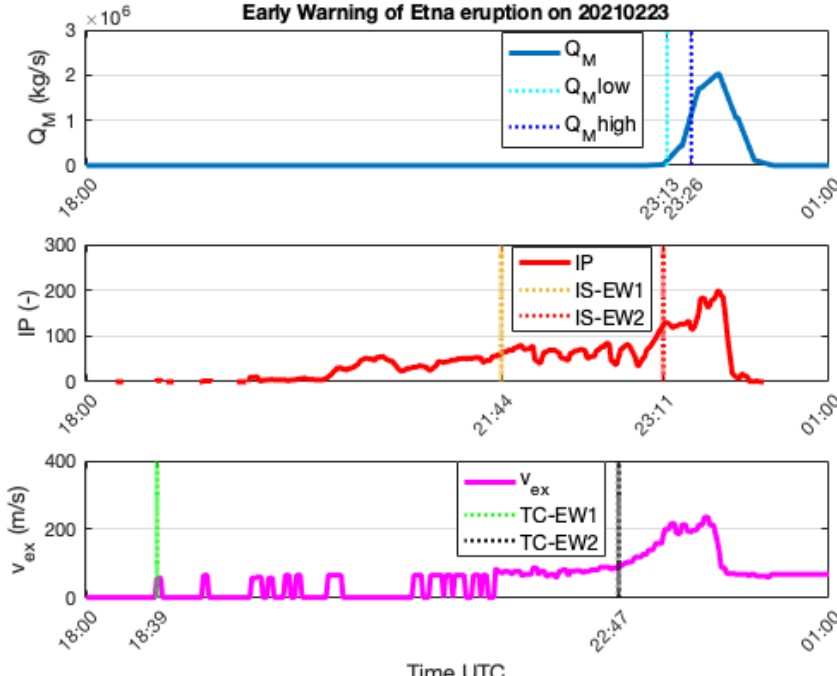

**Figure 14.** Time series of $Q_M$ (kg/s), IP (-), and $v_{ex}$ (m/s) shown as blue, red, and magenta lines, respectively, for the 23 February 2021 Etna lava fountain. The EW level time steps are plotted as vertical dashed lines with different colors, identifying the associated sensors.

## 5. Combined Approach for Paroxysmal Early Warning

According to the previously described results, we developed a theoretical flow diagram of the possible integration of a fully automatic and operative EW system for the identification of Etna PEW (Figure 15). Typically, the EW1 and EW2 level notifications derived from TC and/or IS can be used to inform the WR that an eruption is ongoing.

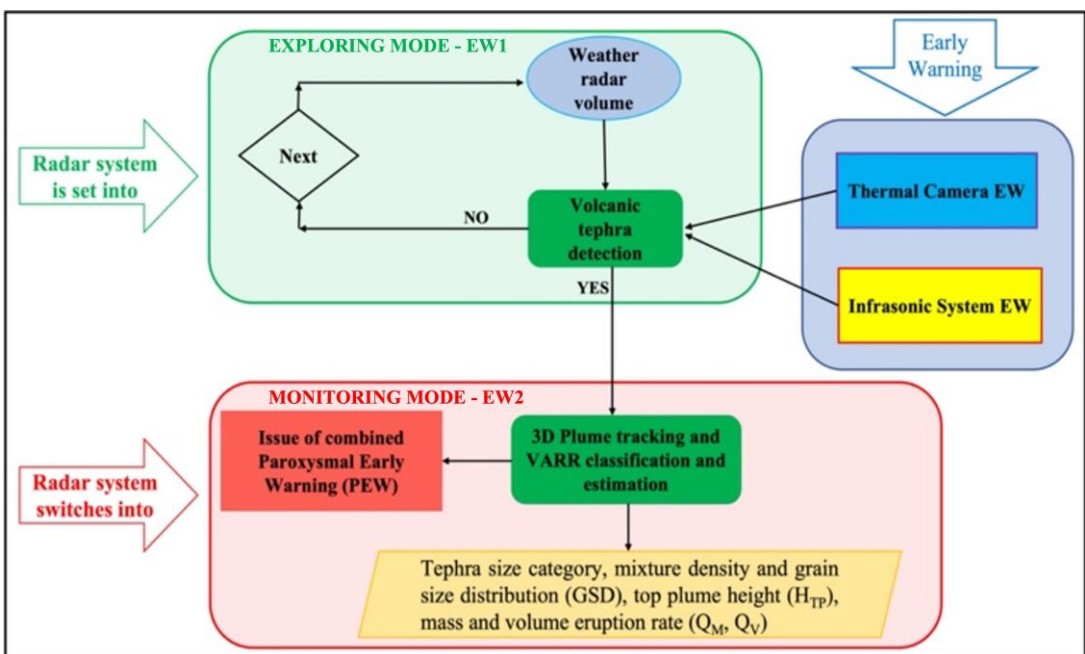

**Figure 15.** Flow diagram of the combined WR, TC, and IS to perform the paroxysmal early warning (PEW). The first EW1 delivered by IS and TC was used, separately or jointly (blue box on the right side), for the WR set to exploring mode (green box) and to activate the area scan. If a volcanic plume was detected (EW2 level), the WR was switched into monitoring mode, activating the 3D plume tracking and detection mode, to quantitatively characterize the main features of the paroxysm.

In this scheme, both the IS and TC (yellow and blue boxes on the right side of Figure 15, respectively), separately or jointly, can be used as a trigger of an automatic procedure of analysis of the WR (detection and tracking of volcanic plume and ESPs retrieval).

The first EW1 signal was used to activate the WR system in exploring mode and run the algorithm to monitor volcanic plumes (green box). In this phase, the WR keeps scanning the 3D surrounding area (azimuth, elevation, and distance). When the volcanic plume was observed (EW2), the WR system was switched into monitoring mode (red box), which detects and retrieves the ESPs (e.g., top plume height ($H_{TP}$), mass eruption rate ($Q_M$), and grain size distribution (GSD)). Regarding these results, we estimated the top plume height ($H_{TP}$ (km)) above the crater and the GSD using a threshold algorithm and applying regressive relations, respectively, on the measured radar reflectivity factor, $Z_{hh}$, of the probed volcanic plume [22,25].

In a few cases, we noted that the TC allowed anticipating the first alert with respect to IS. Hence, the TC is an effective tool compared to IS for anticipating the paroxysm, if the environmental conditions do not compromise the quality of the TC image. However, both the IS and TC are essential to switch the radar to the volcanic plume monitoring mode.

To evaluate the performance of coupling the WR with TC and IS, we computed the alert times as the temporal difference between two equal levels, EWx (where x can be equal to 1 or 2), derived from the IS and TC. In order to test the sensitivity of the WR in detecting the paroxysmal phase, we also combined the EW levels with the WR-derived $Q_M$ levels. We defined the PEW1 lead time, $\Delta t_{PEW1}$, as:

$$\Delta t_{PEW1} = t_{XX-EW1} - t_{YY-EW1/Q_M-low} \qquad (1)$$

where $t_{XX-EW1}$ is the EW1 alert time from the first system (IS or TC) and $t_{YY-EW1}$ is the EW1 alert time from the second system (WR or TC).

Similarly, we defined the PEW2 lead time, considering in this case only the EW2 level, as derived from two systems:

$$\Delta t_{PEW2} = t_{XX-EW2} - t_{YY-EW2/Q_M-high} \tag{2}$$

It is worth noting that PEW2 data are not always available for the selected time windows, so in some cases, the lead time, $\Delta t_{PEW2}$, cannot be properly computed.

The times when TC and IS identified the EW levels and the WR detected the $Q_M$ levels are summarized in Table 1. EW2 and $Q_M$ were not available (NA) for 10 April 2011, 3 and 5 December 2015, and 16 February 2021.

**Table 1.** EW times of levels 1 and 2, as detected by the TC and IS and the WR-$Q_M$ levels (low and high) for the considered Etna explosive eruptions. NA stands for Not Available value.

| Data | WR-QML | WR-QMH | TC-EW1 | TC-EW2 | IS-EW1 | IS-EW2 |
|---|---|---|---|---|---|---|
| **2011-04-10** | 11:25 | NA | 09:04 | 10:18 | 09:38 | 10:00 |
| **2012-04-12** | 14:19 | 14:36 | 11:30 | 12:28 | 11:59 | 13:33 |
| **2013-11-23** | 09:30 | 09:46 | 08:02 | 08:54 | 08:29 | 08:33 |
| **2015-12-03** | 02:01 | 02:23 | 01:44 | 01:53 | 02:38 | NA |
| **2015-12-04** | 08:50 | 09:06 | 08:45 | NA | 07:02 | 09:13 |
| **2015-12-04** | 20:18 | 20:37 | 20:06 | 20:32 | 07:02 | 20:30 |
| **2015-12-05** | 14:51 | NA | 14:30 | 14:40 | 07:02 | 14:46 |
| **2021-02-16** | 16:30 | NA | 16:38 | NA | 16:15 | 16:19 |
| **2021-02-23** | 23:30 | 23:44 | 18:39 | 22:47 | 21:44 | 23:11 |

In the cases where the IS-EW did not provide an alert, such as the lava fountain on 3 December 2015, the TC-EW levels could be a valid complementary warning. On the other hand, in cases where the WR-$Q_M$ levels and TC-EW were missing, due to the particularly low explosive activity ($Q_M$ lower than $10^6$ kg/s), with lower particle sizes and possible partial or total occlusion of the field of view due to meteorological clouds, respectively, the IS-EW could be a complementary warning without any lead time.

The PEW allowed combining the IS with the WR or the TC, as IR-PEW and IT-PEW, respectively, and the TC with the WR, resulting in TR-PEW, as listed in the upper section of Table 2. The TR and IR can be successfully used, jointly or separately, as precursors of the paroxysmal phase, whose tephra plume can be detected by the WR-EW alert in any weather conditions. Moreover, the IS-EW and TC-EW can be effectively used as a trigger for the WR monitoring, with specific and rapid sector scans. The red values listed in Table 2 are related to the lead times of the PEW of both levels, which were anticipated by the WR (IR-PEW and TR-PEW) and the TC (IT-PEW). Only in three cases did the WR preempt the PEW with respect to the first sensors (IS or TC): on 3 December 2015 (IR-PEW1), in the morning on 4 December 2015 (IR-PEW2), and on 16 February 2021 (TR-PEW1). In eight cases, we observed that TC preempted the PEW with respect to the IS: on 10 April 2011, (IT-PEW1), on 12 April 2012 (IT-PEW1 and IT-PEW2), on 23 November 2013 (IT-PEW1), on 3 December 2015 (IT-PEW1), on 5 December 2015 (IT-PEW2), and on 23 February 2021 (IT-PEW1 and IT-PEW2).

Consequently, the TC can be considered as a valid pre-alert system, followed by IS, with similar performance for the selected paroxysms, as long as the former operates in good visibility conditions; conversely, the IS was confirmed as an important pre-alert system.

**Table 2.** Lead times (minutes), $\Delta t$-PEW (1-2), for the considered Etna explosive events: infrasonic system (IS) with respect to the weather radar (WR) (IR_PEW), IS with respect to the thermal camera (TC) (IT_PEW), and TC with respect to the WR (TR_PEW) (upper part of the table). Statistics for each lead time, $\Delta t$-PEW (1-2), in terms of the positive alert (PosAlert), negative alert (NegAlert), and not detected alert (NotAlert), are presented (lower part of the table). NA stands for not available value.

| Data | Lead Time $\Delta t$ (min) | | | | | |
|---|---|---|---|---|---|---|
| | **IR-PEW1** | **IR-PEW2** | **IT-PEW1** | **IT-PEW2** | **TR-PEW1** | **TR-PEW2** |
| **2011-04-10** | 117 | NA | −34 | 18 | 141 | NA |
| **2012-04-12** | 140 | 63 | −29 | −65 | 169 | 128 |
| **2013-11-23** | 61 | 73 | −27 | 21 | 88 | 52 |
| **2015-12-03** | −37 | NA | −54 | NA | 17 | 30 |
| **2015-12-04** | 108 | −7 | 103 | NA | 5 | NA |
| **2015-12-04** | 796 | 7 | 784 | 2 | 12 | 5 |
| **2015-12-05** | 1909 | NA | 1888 | −6 | 21 | NA |
| **2021-02-16** | 15 | NA | 23 | NA | −8 | NA |
| **2021-02-23** | 106 | 33 | −185 | −24 | 291 | 57 |
| | **PosAlert** | | | | | |
| | 88.90 | 44.45 | 44.45 | 33.33 | 88.90 | 55.55 |
| | **NegAlert** | | | | | |
| **Statistic (%)** | 11.10 | 11.10 | 55.55 | 33.33 | 11.10 | NA |
| | **NotAlert** | | | | | |
| | NA | 44.45 | NA | 33.33 | NA | 44.45 |

The statistical values of each PEW were computed as the percentage of positive (PosAlert), negative (NegAlert), and not detected (NotAlert) alerts among all studied cases (lower section of Table 2).

Regarding the IR, we found a rate of 88.90% of PosAlert, with only 11.10% of NegAlert and no NotAlert (NA) for the PEW1, and 44.45% of PosAlert with only 11.10% of NegAlert for the PEW2. There was a rate of 44.45% of NotAlert for the PEW2 due to the lack of the WR-$Q_M$H and IS-EW2. With the IT coupling, we obtained only 44.45% of PosAlert, with 55.55% of NegAlert and no NotAlert detected for PEW1. Instead, for PEW2, each case was identified with an equal rate of 33.33%. The TR coupling showed a rate of 88.90% of PosAlert, 11.10% of NegAlert, and no NotAlert for PEW1, whereas there was a rate of 55.55% of PosAlert without any NegAlert and 44.45% of NotAlert for PEW2.

For IR-PEW1 and TR-PEW1, we obtained a rate of success of 88.90%. This indicates that both systems were reliable in detecting Strombolian activity, while the rate of 55.55% for TR-PEW2 was sufficiently reliable in detecting paroxysmal activity.

It is worth noting that these results could be deduced by applying a Bayesian-weighting approach, where, by combining different sensor measurements, it is possible to define weighted thresholds in a logical framework in order to trigger various levels of alerts, which can better identify different phases of explosive eruptions.

## 6. Conclusions

From the analysis of the data derived from the WR, TC, and IS, we deduced that, for the analyzed Etna eruption events, IR-PEW and TR-PEW anticipated the volcanic plume formation during the lava fountain at least one/two hours prior, while the radar was able to assess the increase in explosive activity, providing both $H_{TP}$ and $Q_M$ in near-real-time. In this analysis, we found that the different PEW1 combinations provided pre-alerts from

a few minutes to about 1 h before the occurrence of the paroxysmal phase, whereas the PEW2 combinations provided pre-alerts within about 1 h.

In particular, the IR-PEW1 was confirmed as a valid pre-alert element, with a rate of 88.90% for positive warnings for the onset of explosive activity, identified as the Strombolian phase. Instead, the TR-PEW2 had a rate of 55.55%, and was suitable to detect the lava fountain phase. Moreover, the IT-PEW1 and IT-PEW2 combinations showed lower percentages of agreement.

It should be noted that these results are from a first analysis of the developed approach, tested on a limited set of Etna eruptive events, which, therefore, need to be confirmed by a more exhaustive analysis on a greater number of paroxysms, also including additional sensors, to make the algorithm even more robust.

The EW results obtained for Etna can be easily applied to other volcanoes where both IS and/or TC and WR co-exist. However, the appropriate thresholds need to be defined for the typical activities of the other volcanoes. In fact, our results indicated that the integration between IS and TC with WR can be efficiently used to define not only the onset of a volcanic eruption, but also to determine key ESPs. We can anticipate that the operational use of the IR-PEW or TR-PEW systems could change the way we monitor volcanoes and largely improve our early-warning systems and near-real-time forecasting in case of explosive eruptions. In addition, these cases provided key insights into the relationship between eruptive intensity, acoustic pressure, and image processing, and some cases may be consistent with a buoyancy-driven ascent for volcanic plumes. All this highlights the importance of integrated IS-WR and TC-WR warning systems. Further studies could help to provide an estimate of the mean buoyant plume velocity field, which is a key factor to forecast the volcanic plume dispersion and to mitigate the risks for air traffic.

**Author Contributions:** Conceptualization, L.M., S.S. and F.S.M.; methodology, L.M., S.S. and F.S.M.; software, L.M.; validation, L.M.; formal analysis, L.M. and F.S.M.; investigation, L.M.; resources, S.S., M.R. and G.L.; data curation, L.M.; writing—original draft preparation, L.M.; writing—review and editing, L.M., S.S., C.B., M.R. and G.L.; visualization, L.M. and S.S.; supervision, S.S. and C.B.; project administration, S.S. and C.B.; funding acquisition, S.S. and C.B. All authors have read and agreed to the published version of the manuscript.

**Funding:** This research was funded by the European Union's Horizon 2020 Research and Innovation Program, under grant agreement No. 731070 (EUROVOLC), and completed in the framework of the INGV Project "Pianeta Dinamico" (D53J19000170001).

**Data Availability Statement:** Data can be available requiring them to authors.

**Acknowledgments:** Infrasonic data are also fully available in the raw format for further development of the early-warning system, whereas data of the microwave weather radar have been kindly provided by the Department of the Italian Civil Protection within the framework of the EUROVOLC project. The thermal camera images are from the monitoring network of Osservatorio Etneo INGV-OE (Catania). We thank all the technicians and technologists for working on its maintenance.

**Conflicts of Interest:** The authors declare no conflict of interest.

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
