# Peer review of "Automatic Early Warning to Derive Eruption Source Parameters of Paroxysmal Activity at Mt. Etna (Italy)"

_remotesensing, doi:10.3390/rs15143501_

Round 1

Reviewer 1 Report

Dear authors, I found your text interesting, even if there are some things to improve. In general, the figures should definitely be improved and a comment should be added to the fact that the system works but for a limited number of events, while it should be tested for a greater number (especially since Etna has shown very frequent activity).

Find my comments in the attached text.

Author Response

We thank the reviewer for his/her useful suggestions. The manuscript has been revised by following all suggestions, improving the algorithm used to analyse the TC images, figures, tables and relative captions and comments. Unfortunately, we don’t have more measurements of other Etna events observed simultaneously by all the sensors. However, we would plan to analyze more events which will occur in future to further validate this approach. We have improved and changed the figures in the paper. All new revisions in the manuscript are highlighted in green

Reviewer 2 Report

Quite a fun paper.  There are only a few minor things I might suggest.  

This is a version of a Bayesian algorithm, where varied data sets are weighted into a logical framework and then criteria are set to trigger various levels of alert.  The good thing is, one can change the weighting based on results, and usually improve the hits and limit the false positives.

I am very pleased that the authors have used the available radar resource from Catania airport.  You might also consider using the free data at Google Earth Engine.  There are some that use that data to look for thermal anomalies (Pergola and others).

The times on the figures have collided and are hard to read when they overlap.  I would space them better.  I would also expand the single sentence figure captions to include the most interesting or important thing about each figure.

Fnially there are a couple references I might include, the first a Ripepe et al from JVGR in in 2002 where they noted the temporal and spatial relationships of infrasound and thermal data, and then a tip of the hat to Matthias Hort and others for early Doppler Radar work from fixed installations at Stromboli.

Thanks again for the chance to review, good luck!

First, make use of the word "forecast".  This is a good word for what you're trying to do.  "Predict" or some of the other words are a little awkward.  Another word to use, "alert" instead of "alarm".  As far as the English goes, it is fine, perhaps a quick double check would help, the Microsoft editor helps me a lot.

Author Response

We very thank the reviewer for all the helpful suggestions. The manuscript has been revised following all suggestions to corroborate the main text, adding new references, improving figures, tables and their respective captions. It’s true, this is similar a Bayesian approach, where we combine observations of three different sensors and define weighted thresholds in a logical framework, to identify the different stages of explosive eruptions. As the reviewer rightly notes, by changing the weighting based on results, we can improve hits and limit false positives. In this first work we only wanted to test the validity of this approach on a limited set of Etna explosive events, but we plane to analyze more events to further validate this approach. We have improved and changed the figures in the paper. All new modifications in the manuscript are highlighted in green.

Reviewer 3 Report

This manuscript is valuable for publishing on Remote Sensing. Proposed method is combination of infrasound, thermal activity and detection of plume using weather radar. This method covers all the instrumentation to detect volcanic plume and is helpful to detect similar volcanic eruption on the other volcanoes in the world.

I would like to show some comments.

Major comments

1) The authors detected EW1 and EW2 from IP, vex and QM for 9 paroxysmal events. If the automatic EW method is continously operated for real time data, it is possible for IS and TC to issue incorrect EW. How about the rate of incorrect EW? Or, what kind of condition lead the EW system to incorrect EW?     

2) Please decribe the method to estimate exit velocity vex, QM and GSD. Considering the title of this manuscript, quatification of these parameters is important.

3) PosAlerts are obtained from 88.9% of IR_PEW1. This indicates the IR_PEW1 is relaiable, suggesting transition of eruptive activity from Strombolian eruption to paroxysmal activity. Would you like to add some discussion on the transition of the eruptions from volcanolgoical aspect?

Minor comments

L113

It is better to describe the sensor type.

Figure 1

Explanation of a) and b) is necessary.

Whats are "ARRAY 1,2,3,4". Is this an enlargement of MTV and ETN?

L128

Please describe the detection algorithm for number per minute.

Figure 2

When no paroxyms detected, IP=-1. Why is IP minus? IP is the product of amplitude x number.

L151

It is better to describe the type of thermal camera and measurement range of temperature.

L157

How do you estimate exit velocity from the height of the IJR? Please decribe the method.

Figure 5

I cannot well understand the meaning of right axis IP(-)/vex (m/s). The right axis is common for IP and vex? IP has unit in Pa and vex has unit in m/s. Did IS-EW1 occur at the same time as TC-EW1?

L299

The value of vex seems small er than 190m/s. Around 160?

L300

Is "greater" correct? "smaller"?

L301

IP signal reached >500 immediately before 17:00.

L344

This is an importnt point considering the title of this manuscript. How do you estimate QM and GSD using weather radar data? Please describe the method for estimation.

Figure 14

What does it mean "Weather radar volume"? This may indicate tracking mode. Is this 2D tracking?

L366

Is the symbol "tyy-EW1" correct, considering equation (1)?

L397

Table II -> Table 2

L408-413

Decimal points are indicated by "." not ","

Table 2

Unit of Dt is minute?

Unit of statistics is %?

Decimal points are indicated by "." not ","

L434

66,7% -> 66.7%

Author Response

We thank the reviewer for the useful suggestions. The manuscript has been revised by following all suggestions. We have improved and changed the figure in the paper. Please, find hereafter our replies to all your comments in red, whereas all modifications in the main text are highlighted in green.

Reviewer 4 Report

This manuscript presents an automatic early warning system for Etna volcano based on infrasonic, thermal camera and weather radar monitoring. It is a very relevant topic, based on the monitoring experience of the different systems at Etna, and integrating these methods in a systematic method. It definitely deserves to be published in Remote Sensing after minor revision.

In the attached file, I included minor comments and corrections. I recommend a final check of English Language to correct typos, grammatical and drafting errors.

Author Response

We thank the reviewer for his/her useful suggestions. The manuscript has been revised by following all suggestions, improving the algorithm used to analyse the TC images, figures, tables and relative captions and comments. All new revisions in the manuscript are highlighted in green.

Round 2

Reviewer 1 Report

Dear Authors, in my opinion, the revised text has shown improvement and can be accepted in its current form.